# Organization of high-level visual cortex in human infants

Ben Deen[1], Hilary Richardson[1], Daniel D. Dilks[1,2], Atsushi Takahashi[1], Boris Keil[3,4], Lawrence L. Wald[3,5], Nancy Kanwisher[1] & Rebecca Saxe[1]

How much of the structure of the human mind and brain is already specified at birth, and how much arises from experience? In this article, we consider the test case of extrastriate visual cortex, where a highly systematic functional organization is present in virtually every normal adult, including regions preferring behaviourally significant stimulus categories, such as faces, bodies, and scenes. Novel methods were developed to scan awake infants with fMRI, while they viewed multiple categories of visual stimuli. Here we report that the visual cortex of 4–6-month-old infants contains regions that respond preferentially to abstract categories (faces and scenes), with a spatial organization similar to adults. However, precise response profiles and patterns of activity across multiple visual categories differ between infants and adults. These results demonstrate that the large-scale organization of category preferences in visual cortex is adult-like within a few months after birth, but is subsequently refined through development.

[1] Department of Brain and Cognitive Sciences and McGovern Institute, Massachusetts Institute of Technology, Cambridge, Massachusetts 02139, USA. [2] Department of Psychology, Emory University, Atlanta, Georgia 30322, USA. [3] Athinoula A. Martinos Center for Biomedical Imaging, Department of Radiology, Harvard Medical School, Massachusetts General Hospital, Charlestown, Massachusetts 02129, USA. [4] Institute of Medical Physics and Radiation Protection, Department of Life Science Engineering, Mittelhessen University of Applied Science, Giessen 35390, Germany. [5] Harvard-MIT Division of Health Sciences and Technology, Massachusetts Institute of Technology, Cambridge, Massachusetts 02139, USA. Correspondence and requests for materials should be addressed to B.D. (email: benjamin.deen@gmail.com).

In human adults, the cortex shows a systematic spatial and functional organization. Responses in visual cortex are driven by high-level, behaviourally relevant categories, including human faces, bodies, objects and natural scenes, both within circumscribed, highly selective regions[1–4], and in graded response patterns across larger swaths of cortex[5–7]. The origins of these responses have been the topic of intense debate: are they learned, reflecting a gradual accrual of expertise, or do they reflect innate predispositions?

A key constraint on theories of cortical development would be evidence of when these responses emerge in cortex. However, the functional organization of high-level responses in visual cortex has never been tested in infants, and existing indirect evidence makes contradictory predictions. Slow, hierarchical development of visual functions over years is suggested by late developmental change in children aged 4–10 years[8,9], slow and staggered time courses of myelination[10] and cortical thinning[11], and late developmental change in juvenile macaques[12,13]. By contrast, early functional maturation of cortex in infancy is consistent with high-level responses measured by electroencephalography (EEG)[14,15] and near-infrared spectroscopy (NIRS)[16,17], rare electrophysiological recordings from infant macaques[18], and the sophisticated cognition of pre-verbal infants revealed by the modern developmental psychology[19].

The main obstacle to resolving this debate is the difficulty of neuroimaging awake infants. The imaging techniques most commonly used in human infants (EEG and fNIRS) lack the coverage and resolution needed to measure the spatial organization of cortex. Only two prior studies have collected functional magnetic resonance imaging (fMRI) data from awake infants, and because of infants' limited tolerance, it has been difficult to collect sufficient data to test replicability or functional profiles of response[20,21]. Here we implement novel methods for awake infant fMRI to study the early development of high-level visual responses in cortex. We employ a number of technical advances to increase participant comfort, optimize signal strength and minimize head motion artefacts: (1) infant-sized MR head coils; (2) quiet pulse sequences; (3) dynamic and engaging visual stimuli; and (4) a combination of extant and novel data analysis techniques for minimizing motion artefacts.

Our data demonstrate that by 4–6 months of age, human infants have category-sensitive visual responses to faces and scenes, with a spatial organization mimicking that observed in adults. However, we also observe differences: both in response profiles across multiple categories (which were less selective in infants), and in patterns of response across cortex. Thus, the overall functional organization of high-level visual cortex develops very early, and is subsequently refined.

## Results

**fMRI findings**. We obtained low-motion fMRI data from 9 infants (of 17 tested; age 3–8 months; Supplementary Table 1), while they viewed engaging, brightly coloured, infant-friendly movies of faces, natural scenes, scrambled scenes, human bodies and objects (Supplementary Fig. 1). We first compared responses to faces versus scenes, because in adults this comparison yields the most robust differential responses, and delineates a large-scale spatial organization of extrastriate cortex[22,23]. Face- or scene-preferring regions in occipitotemporal cortex were observed in eight of nine infants, with a similar spatial organization as in adults (Fig. 1; Supplementary Figs 2 and 3). In individual infants, face-preferring regions were observed in the fusiform gyrus, lateral occipital cortex, superior temporal sulcus (STS) and medial prefrontal cortex; scene-preferring regions were observed in the parahippocampal gyrus and lateral occipital cortex. Many of

these regions showed reliable responses in a group analysis, demonstrating generalization across infants (Fig. 1). Region-of-interest (ROI) analyses corroborated whole-brain results, demonstrating reliable face and scene preferences in data independent from those used to define ROIs, in all regions tested (Fig. 2; Supplementary Figs 4–7; Expt. 1, $n = 9$, permutation test; ventral face region, $z = 2.85$, $P = 2.2 \times 10^{-3}$ lateral face region, $z = 3.27$, $P = 5.4 \times 10^{-4}$; STS face region, $z = 4.74$, $P = 1.1 \times 10^{-6}$; ventral scene region, $z = 6.41$, $P = 7.3 \times 10^{-11}$; and lateral scene region, $z = 3.43$, $P = 3.0 \times 10^{-4}$). In six infants who participated in more than one experiment, these preferences were also replicated using distinct face and scene movies (Expts. 2–8, $n = 6$, permutation test; ventral face region, $z = 2.22$, $P = 0.013$; lateral face region, $z = 2.51$, $P = 6.0 \times 10^{-3}$; STS face region, $z = 4.19$, $P = 1.4 \times 10^{-5}$; ventral scene region, $z = 5.64$, $P = 8.5 \times 10^{-9}$; and lateral scene region, $z = 5.00$, $P = 2.9 \times 10^{-7}$).

These results demonstrate that the spatial organization of preferential responses to faces versus scenes is similar in 4–6-month-old infants and in adults, extending throughout the ventral visual stream and even into prefrontal cortex. In subsequent analyses, we sought to constrain the functional interpretation of these responses. Are cortical regions in infants responding to highly specific visual categories[1,2,4], to broader visual or semantic dimensions[5,6], or to lower-level visual features that co-vary with high-level categories[24–27]? Do large-scale patterns of response to categories other than faces and scenes change over development? Measuring responses to multiple visual categories enabled us to ask these questions.

Do preferential responses to faces and scenes in infants reflect a high-level category preference, or a bias toward lower-level visual features, such as eccentricity, spatial frequency or rectilinearity (the presence of 90° angles)[24–27]? We tested whether cortical responses in infants were better predicted by these lower-level visual features than by high-level categories. In Experiment 2, scenes and scrambled scenes were reduced to 80% the size of face and body movies, but category preferences were unaffected, suggesting that these responses were not driven by eccentricity (Supplementary Fig. 7B). Across all experiments, rectilinearity and spatial frequency content of the movies predicted responses no better, and in scene regions significantly worse than modulation by visual category (Fig. 3). The visual category model was particularly better for scene-preferring regions because the control condition in most experiments (scrambled scenes) had high spatial frequency and high rectilinearity, most clearly differentiating the predictions of the lower-level features from the visual category model. For face-preferring regions, category and low-level feature models made similar predictions for these stimuli; future experiments including a low-spatial-frequency, highly-curvilinear control condition will clarify the responses of these regions. Overall, however, our data suggest that by 4–6 months, category-sensitive cortical responses are not primarily driven by lower-level visual features.

Another outstanding question is whether responses to faces and scenes in infants reflect regions with highly selective responses to specific categories, or weaker-graded preferences across multiple categories. In adults, for example, the fusiform face area and parahippocampal place area have highly selective response profiles, preferring faces or scenes to any other visual category (for example, objects, bodies, animals, foods and so on)[2,4], whereas broad areas around these regions have graded preferences predicted by coarser semantic dimensions[5,6]. We searched for highly selective regions by contrasting faces (or scenes) to objects. In adults, these contrasts revealed the predicted spatially focal, strongly selective regions: each region

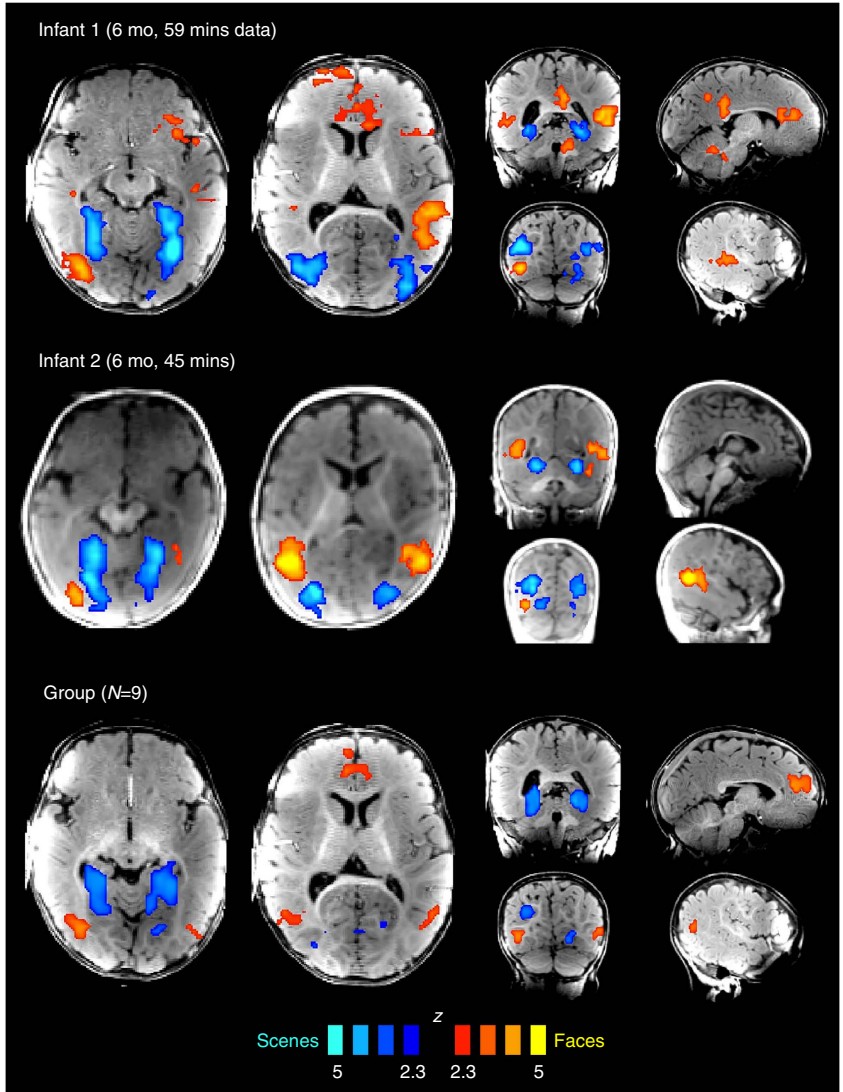

**Figure 1 | Category-sensitive responses to faces and scenes in infants show adult-like spatial organization.** Regions preferring faces over scenes are reported in red/yellow, and regions preferring scenes over faces in blue. The top two rows of whole-brain activation maps show results from the two individual infants with the largest amount of usable data, while the third shows a group map with statistics across infants. Maps are thresholded at $P < 0.01$ voxelwise, and corrected for multiple comparisons using a clusterwise threshold of $P < 0.05$.

showed a higher response to its preferred category than to all three other categories (Fig. 4; Supplementary Fig. 3; permutation test comparing faces or scenes to objects, $n = 3$; ventral face region, $z = 5.54$, $P = 1.5 \times 10^{-8}$; lateral face region, $z = 4.35$, $P = 6.8 \times 10^{-6}$; STS face region, $z = 7.13$, $P = 5.0 \times 10^{-13}$; ventral scene region, $z = 6.10$, $P = 5.3 \times 10^{-10}$; and lateral scene region, $z = 5.12$, $P = 1.5 \times 10^{-7}$). In infants, however, no region showed a higher response to faces or scenes over objects (permutation test, $n = 6$; ventral face region, $z = -0.75$, $P = 0.77$; lateral face region, $z = 0.91$, $P = 0.18$; STS face region, $z = 1.40$, $P = 0.08$; ventral scene region, $z = -1.36$, $P = 0.91$; and lateral scene region, $z = 0.81$, $P = 0.21$). Similar results were obtained for a range of ROI sizes (Fig. 4): adults showed a significant response to faces (or scenes) over objects for all regions and ROI sizes (permutation test, $n = 3$, all $P$'s $< 0.05$), while infants did not show a significant response for any region or ROI size, including ROIs as small as $0.8\,cm^3$ (permutation test, $n = 6$, all $P$'s $> 0.05$). Thus, within the spatial resolution of our methods, we find no evidence that the difference between groups reflects a change in the size of selective regions.

Could these null findings result simply from poor data quality in infants? Several observations argue against this interpretation. First, standard errors did not differ substantially across infants and adults, and when a reduced subset of adult data was analysed to inflate standard errors, the same results were obtained (Supplementary Fig. 8). Second, although no region preferred faces (or scenes) to objects in infants, the reverse contrast in exactly the same data revealed robust responses to objects, compared with either faces or scenes, with adult-like spatial organization in temporal and parietal cortex (Supplementary Fig. 9). Thus, while the large-scale spatial organization of responses to faces versus scenes is present in infants and remains a principal dimension of cortical organization into adulthood, highly selective regions for particular categories apparently emerge later in development, perhaps requiring more extensive visual experience.

In addition to the absence of category-selective regions, we found evidence for developmental change in the large-scale patterns of functional response across multiple categories. To summarize and quantify the spatial structure of responses

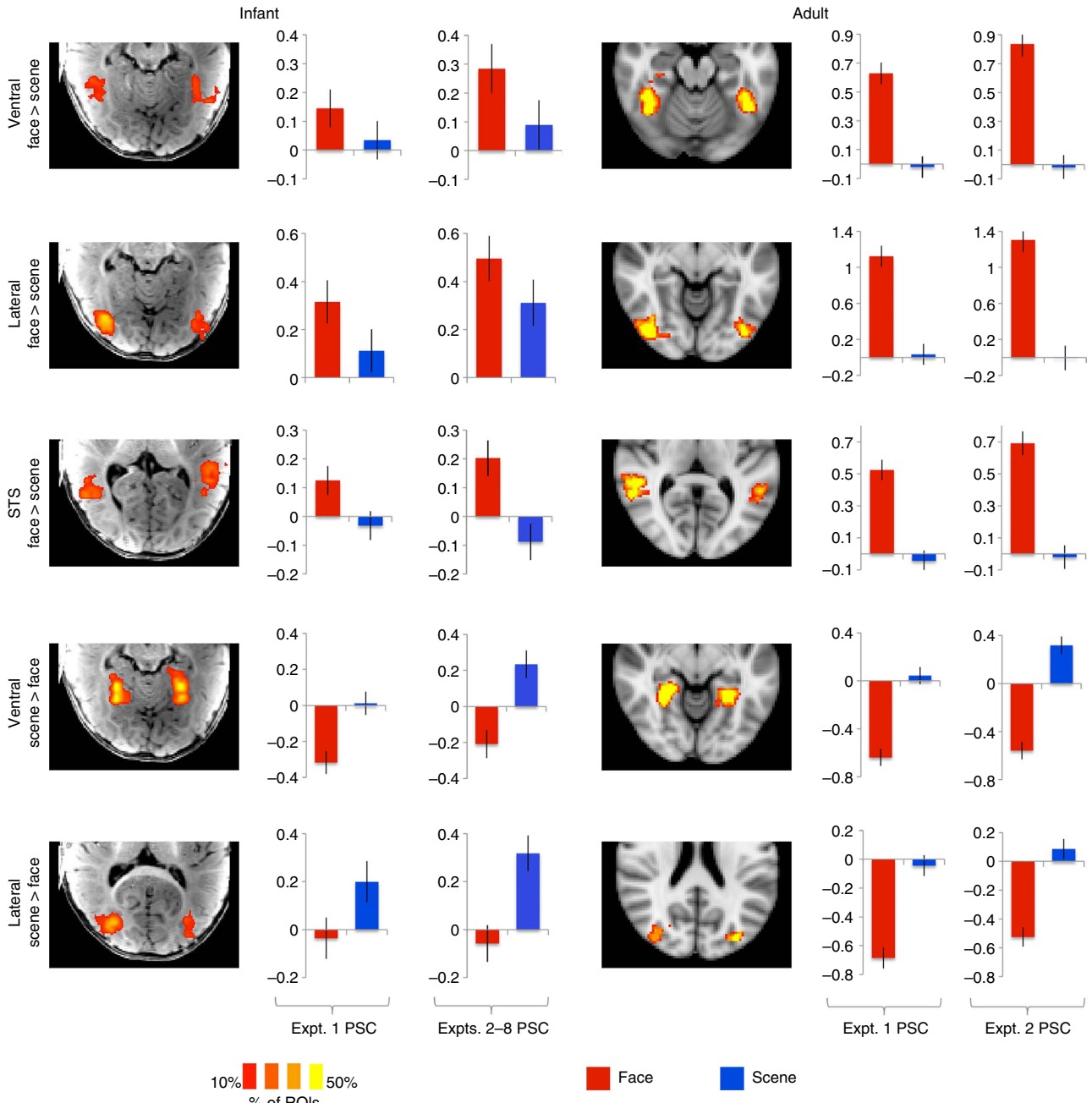

**Figure 2 | The location and reliability of responses to faces and scenes is consistent across infants and adults.** Brain images show heat maps of region-of-interest (ROI) locations across participants (% of ROIs that included a given voxel), with ROIs defined as the top 5% of voxels responding to faces over scenes (or vice versa) within an anatomical region. Bar plots show each ROI's response (per cent signal change, PSC) to faces and scenes in independent data, separately for Expts. 1, 2–8. Error bars show the standard deviation of a permutation-based null distribution for the corresponding value. Baseline corresponds to the response to scrambled scenes (Expts. 1–3, 7–8) or scrambled objects (Expts. 4–6). Statistics for infant data are presented in the main text; as expected, face and scene preferences were highly significant in adults for all regions (permutation test, $n = 3$; all $P$'s $< 10^{-15}$).

to multiple categories, we computed representational similarity matrices, capturing the similarity of spatial patterns of response across categories[28]. While face and scene responses were dissimilar in both groups, consistent with the results above, the pattern of similarity across all categories differed between infants and adults (Fig. 5; Supplementary Fig. 10). Representational similarity matrices across the four categories were highly similar within adults ($n = 3$, mean Kendall's tau $= 0.91$), and moderately similar within infants ($n = 6$, mean Kendall's tau $= 0.41$), but dissimilar between groups

(mean Kendall's tau $= 0.14$; significantly lower than within group similarity for both infants, $P = 0.024$, and adults, $P = 0.012$, permutation test). Thus visual responses to multiple categories differ in infants and adults, as measured both by response profiles of focal regions, and distributed patterns of response across cortex.

## Discussion

Using novel methods to acquire and analyse fMRI data from awake human infants, this study demonstrates that the cortex of

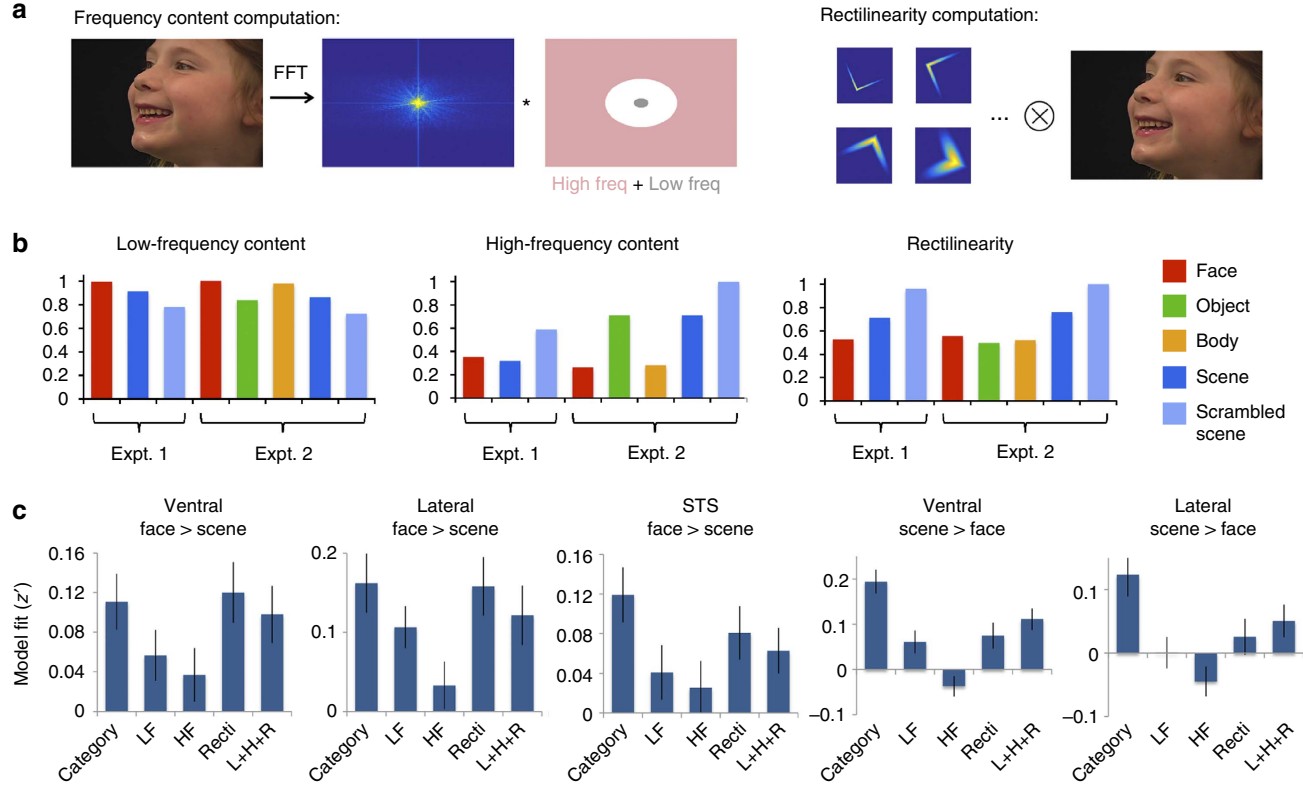

**Figure 3 | Comparison of categorical and visual feature-based models of region-of-interest (ROI) responses.** (**a**) Schematic showing how high- and low-frequency content and rectilinearity were computed from movie frames. (**b**) Mean values of these visual features across the stimuli used in Expts. 1–2, normalized such that the maximum value across categories is set to 1. (**c**) Model fits of category and visual feature models to ROI responses, with error bars specifying standard error. In all three face-preferring regions, there was no significant difference between the category model and the best-performing visual feature model (ventral face region, $t(54) = -0.48$, $P = 0.64$; lateral face region, $t(54) = 0.25$, $P = 0.80$; STS face region, $t(54) = 1.55$, $P = 0.13$). In these regions, the category model (including a high response to faces) and the rectinilinearity model (a low response to rectilinearity) made very similar predictions; other types of stimuli (such as curve-scrambled faces) may be needed to distinguish these hypotheses. In contrast, for scene regions, the category model and low-level feature models made distinct predictions due to the inclusion of a highly rectilinear non-scene condition (grid-scrambled movies). For the two scene-preferring regions, the category model significantly outperformed all visual feature models. For brevity, we report statistics only for the comparison with the best-performing model (ventral scene region, $t(54) = 3.56$, $P = 7.8 \times 10^{-4}$; lateral scene region, $t(54) = 2.56$, $P = 0.013$). HF, high-frequency content; $L+H+R$ = low-frequency content, high-frequency content and rectilinearity; LF, low-frequency content; Recti, rectilinearity.

4–6-month-old human infants is already spatially organized, with distinct regions responding preferentially to human faces versus natural scenes. The spatial structure of these responses is very similar to that observed in adults, and extends throughout cortex, including occipital, temporal, parietal and frontal regions. Thus, while the anatomical maturation of human cortex is slow and asynchronous, basic aspects of functional organization are present across cortex from a very early age.

Prior fMRI studies have observed category-sensitive responses in high-level visual cortex in children as young as 4 years[8]. By demonstrating that these responses exist by 4–6 months of age, the current study provides a stronger constraint on theories of cortical development: this functional organization must either be determined innately, without any need for visual experience, or develop within the first few months of life. A limited role for visual experience in the development of category-sensitive responses is consistent with evidence that in congenitally blind adults, category-sensitive responses in visual cortex develop in the absence of any visual input[29,30].

The observation of face-sensitive functional responses in human infants is also consistent with prior evidence from EEG and NIRS[14–17]. Using fMRI, our results go beyond those prior studies because we are able to assess the precise spatial organization of category-sensitive responses, and to measure responses in non-superficial regions, such as ventral temporal cortex. This novel evidence of the functional organization of cortex in infancy can be directly related to the extensive fMRI literature on visual responses in adults. In addition to providing spatial resolution, the current data provide better functional characterization of cortical responses in infants. By acquiring a large amount of high-quality data within individual infants, we are able to measure responses to multiple categories, and to internally replicate our finding of face and scene responses, across experiments that used different specific movie stimuli. We also provide initial evidence that infants' responses to high-level, behaviourally significant categories cannot be explained in terms of responses to simple lower-level visual features.

While our data indicate that the spatial organization of responses to faces and scenes is remarkably adult-like, we additionally observed that both the fine-grained selectivity and spatial pattern of activity across multiple categories change with age. In particular, and in contrast to adults, infants did not have strongly category-selective regions, that is, circumscribed regions showing a robustly stronger response to one category than to any other. Differences between infants and adults must be interpreted with caution, given the marked differences in brain size and general visual and cognitive function.

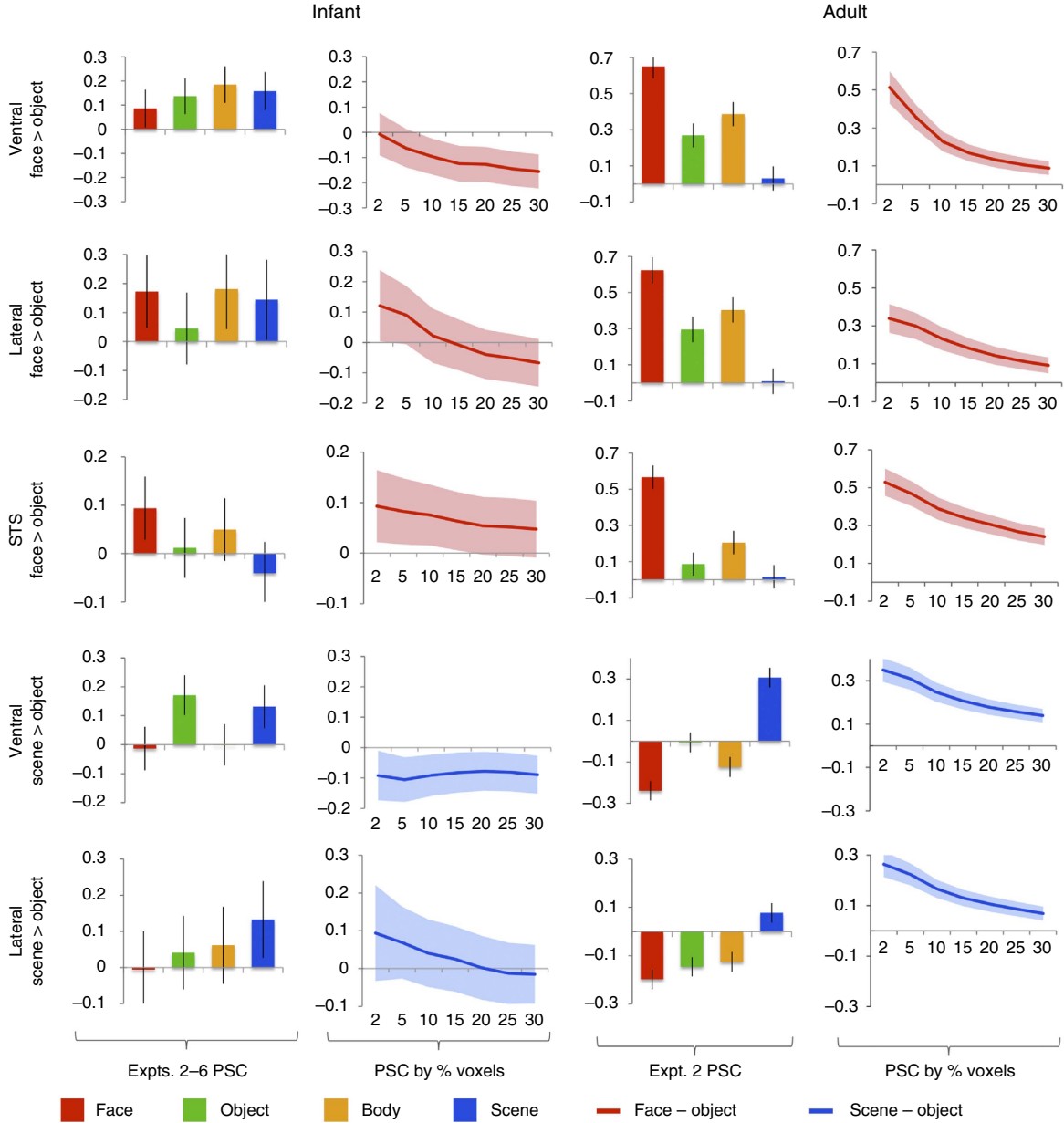

**Figure 4 | Infants lack strongly category-selective regions.** Region-of-interest (ROI) responses (per cent signal change, PSC, in independent data) in regions defined by comparing faces to objects and scenes to objects, in infants and adults. Bar plots show responses of ROIs defined as the top 5% of voxels within an anatomical region, while line graphs show how the difference between face and object or scene and object responses varies as a function of ROI size. Adults show strongly selective responses, substantially higher to the preferred category than any other category, while infants do not show a reliable or selective response at any ROI size. Error bars show the standard deviation of a permutation-based null distribution for the corresponding PSC value or PSC difference. Baseline corresponds to the response to scrambled scenes (Expts. 2–3) or scrambled objects (Expts. 4–6).

For instance, one possibility is that in adults, category-selective responses are enhanced by top-down feedback and selective attention, which are not yet mature in infants. Nevertheless, these data are consistent with the hypothesis that the early-developing large-scale functional organization of category preferences in cortex provides a scaffolding for subsequent refinement of responses, leading ultimately to the strongly category-selective regions observed in adults[31]. The process of refinement likely depends on both physiological maturation (for example, myelination of long-range connections between brain regions) and visual experience. For example, the visual word form area develops as a result of experience with a specific orthography[32], but is guided by pre-existing patterns of anatomical connectivity[33]. Similarly, extensive training with novel symbols can generate selective responses in a cortical region in macaques; the location of this region is consistent across animals, suggesting refinement based on a pre-existing scaffold[12,13].

These results point to myriad future questions, including: what is the time course of the development of category-selective visual regions during and after the first year of life? How do maturation and visual experience interact to drive this development? And does a similar principle (an initial preference that is subsequently refined) apply to the development of functionally specific regions in other perceptual and cognitive domains? We hope that the methods introduced here will aid in future investigations of these questions.

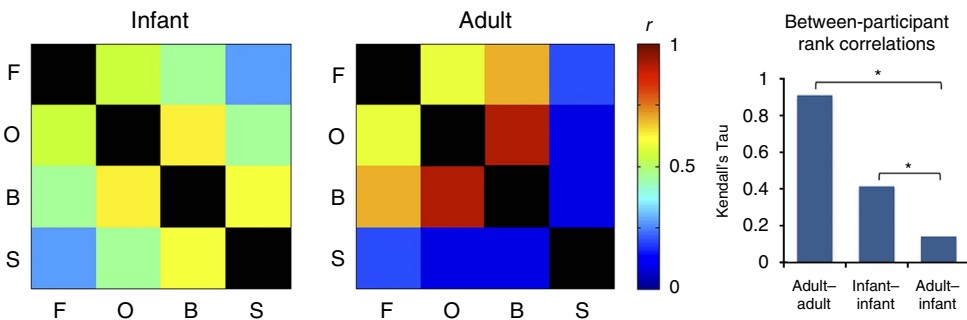

**Figure 5 | Distinct representational similarity for multiple visual categories in infants and adults.** Left two images show representational similarity matrices: correlations between spatial patterns of response across extrastriate visual cortex, to faces (F), objects (O), bodies (B) and scenes (S). Bar graph on the right shows rank correlations (Kendall's tau) between similarity measures from pairs of participants. Within group (adult–adult and infant–infant) rank correlations are significantly higher than between group (adult–infant) rank correlations, indicating a reliably distinct similarity structure across groups. *denotes $P < 0.05$.

## Methods

**Participants.** We scanned 17 infants (age 2.3–8.6 months, three female) and acquired useable (low-motion) data from nine infants (age 3.0–8.0 months, one female). We also scanned three adults (age 27–34 years, one female) for comparison (Supplementary Table 1). Because low-motion data from infants was relatively rare, whenever possible we scanned infants in multiple sessions (between 1 and 16 scan sessions per infant, for a total of 63 sessions). Sessions occurring within a month were analysed together as a single data set; sessions separated by more than a month were analysed as separate data sets (this occurred for five infants; only one data set per infant was used for group analyses). Adult participants and parents of infant participants provided written, informed consent, as approved by the Committee on the Use of Humans as Experimental Subjects at MIT.

**Paradigm.** Stimuli were infant-friendly dynamic movie clips depicting faces, objects, bodies, and scenes (Supplementary Fig. 1). Participants initially viewed Experiment 1 (Expt. 1), a two-condition (face, scene) experiment, with grid-scrambled scenes included as a baseline (pilot testing indicated that infants would not tolerate a baseline with less visual structure). When time permitted, we additionally ran Experiment 2 (Expt. 2), a four-condition (face, object, body and scene) version, with distinct face and scene movies, a scrambled scene baseline, and both scene and scrambled scene movies presented at 80% size, to minimize the possibility of a retinotopic confound in the scene versus face comparison. In certain cases, experiments (Expts. 3–8) using different movies of the same categories were used, to further test generalization of responses across specific movies; these experiments, as well as more detail on stimuli, are further described below (Further paradigm details). Stimuli were presented in 18 s-long blocks, typically comprising six 3 s-long movie clips. Baseline blocks occurred every seven blocks (Expt. 1) or five blocks (Expt. 2); experimental blocks were ordered pseudorandomly between baseline blocks. During infant functional scans, an experimenter or parent lay in the scanner bore to monitor the infant, and told the experimenters if the infant closed his or her eyes, fell asleep or fussed out. For infants, individual runs were not fixed in duration, but instead ended whenever the infant fussed out or fell asleep. For adults, runs lasted 22 blocks (Expt. 1) or 21 blocks (Expt. 2), with a baseline block at the start and end of each run. Adults received five runs each of Expt. 1 and Expt. 2. Parents of actors in stimulus videos provided written, informed consent for the publication of images in Figure 3 and Supplementary Figure 1.

**Data acquisition.** MRI data were acquired using a Siemens 3T MAGNETOM Tim Trio scanner (Siemens AG, Healthcare, Erlangen, Germany). We used a standard 32-channel head coil for adult participants, and a custom-built infant-sized 32-channel head coil for infants[34]. The latter was shaped like a reclined car seat to increase comfort, and had coil elements close to the infant's head, to reduce head motion and increase signal-to-noise ratio. For infants whose heads did not fit in this coil, a 32-channel head coil designed for 5 year olds was used instead. To further increase infant comfort, we acquired data using a quiet (70–72 dB sound pressure level) T2*-weighted pulse sequence[35], sensitive to blood-oxygen-level-dependent contrast (repetition time (TR) = 3 s, echo time (TE) = 43 ms, α = 90°, field of view (FOV) = 192 mm, matrix = 64 × 64, slice thickness = 3 mm, slice gap = 0.6 mm). For infants, we used 18–24 near-axial slices, using the minimum number of slices required to cover occipitotemporal cortex for a given head size, because pulse sequence audio volume scaled with number of slices; for adults, we used 36 near-axial slices for whole-brain coverage. Infants were swaddled during all scans to reduce body movement.

Anatomical images were only collected in certain cases, because our focus was normally to collect as much awake functional data as possible, and because collecting a high-quality anatomical typically required the infant to be asleep to reduce motion. When anatomicals were collected, we used one of three T1-weighted pulse sequences of varying length, using briefer, lower-quality sequences when an infant would only stay still for a short duration. These included a 24 s sequence (TR = 283 ms, TE = 1.8 ms, flip angle α = 9°, FOV = 159 mm, matrix = 106 × 106, slice thickness = 1.5 mm, 96 sagittal slices), a 2.2-min sequence (TR = 800 ms, TE = 3.43 ms, flip angle α = 9°, FOV = 160 mm, matrix = 160 × 160, slice thickness = 1 mm, 144 sagittal slices), and a 6.5-min sequence (TR = 2530 ms, TE = 1.64 ms, flip angle α = 7°, FOV = 256 mm, matrix = 256 × 256, slice thickness = 1 mm, 176 sagittal slices, acceleration factor = 2, 24 reference lines). In adults anatomicals were acquired using the 6.5-min sequence.

**Data selection.** Data were processed primarily using custom scripts, with tools from the FMRIB Software Library (FSL) version 4.1.8 and Freesurfer additionally used for registration and motion correction. Because some of our infant data contained a substantial amount of head movement, and because head motion causes highly deleterious artefacts in fMRI data[36], we first aimed to discard high-motion data that could corrupt our results and lead to false negatives. Each run was first motion corrected by registering each volume to the middle volume, using rigid transformations determined by FSL's MCFLIRT. Using the motion parameters estimated by this correction, we applied a technique known as scrubbing[37,38], removing pairs of adjacent volumes with >0.5 mm of total translation or 0.5° of total rotation between them. We also removed volumes where the participant's eyes were closed, and the first three volumes of each run (to allow the MR system to equilibrate).

While this technique is effective in removing artefactual spikes of response that occur at high-motion time points, it can still leave large baseline shifts in voxels' time courses that occur when a participant's head moves substantially and remains in a new location relative to the head coil and external magnetic field. We thus instituted a second cutoff on scrubbed data, at pairs of adjacent volumes with >2 mm of total translation or 2° of total rotation between them. At these cutoff points, we temporally split runs to form 'pseudoruns' of scrubbed data, where the head was in a relatively consistent position. These pseudoruns were subsequently analysed as one would normally analyse a full run. Pseudoruns were kept for analysis if they contained at least 24 time points, as well as six time points per condition for all conditions (where condition timing was lagged by 6 s to account for hemodynamic delay), such that responses to each condition could be estimated. Last, participants were included in analyses if they had at least 5 min of data saved after this procedure, across experiments.

Supplementary Table 1 shows the amount of data acquired and saved, across participants. We initially acquired 23.06 h of data across 17 infants, and were left with 4.26 h of data across 9 infants after motion screening. Resulting pseudoruns in infants ranged in length from 1.2–17.5 min (mean 4.3 min). While this procedure led to a substantial reduction in data quantity, it drastically reduced the amount of head motion present in the resulting data, reducing mean volume-to-volume translation from 1.11 to 0.13 mm, and mean rotation from 1.69° to 0.17°. In adults, neither scrubbing nor pseudorun selection resulted in any volumes being removed, such that pseudoruns were equivalent to the original runs. Adult data had mean volume-to-volume translation of 0.04 mm, and mean rotation of 0.02°.

**Data preprocessing.** Pseudoruns were first motion-corrected by registering each volume to the middle volume, using rigid transformations determined by FSL's MCFLIRT. Data were skull-stripped using FSL's Brain Extraction Tool, and spatially smoothed using a 3 mm-full-width at half-maximum Gaussian kernel.

**Data registration.** To combine data across pseudoruns, middle volumes from each pseudorun for a given participant were all registered to a common target middle volume, chosen to have minimal distortion. All registration was performed using FSL's FLIRT, unless otherwise noted. In infant data, head motion across pseudoruns posed challenges for this registration: different volumes could have different positions within the bounding box, and different types of nonrigid distortion. To optimize registration, we thus adopted the following procedure: (1) middle volumes were algorithmically registered to target volumes using both a rigid transformation and a general affine transformation; (2) translation and rotation parameters for both of these transformations were hand-tuned to improve registration quality; and (3) we selected whichever resulting transformation (hand-tuned rigid or hand-tuned affine) provided a more accurate registration based on visual inspection of anatomical landmarks. For adult data, middle volumes were registered to the target using a rigid transformation.

For infant data, in cases where anatomicals were collected, target functional volumes were registered to anatomical images using a rigid transformation, with translation and rotation parameters subsequently hand-tuned. For adult data, because surface reconstructions could be obtained, target functionals were registered to anatomicals with a rigid transformation determined by Freesurfer's bbregister. Anatomical images in adults were in turn registered to the Montreal Neurological Institute (MNI) 152 template brain using a nonlinear transformation determined by FSL's FNIRT.

Last, we aimed to register data across infants, for the purposes of registering search spaces for ROI analyses (described below), and to compute group-level whole-brain statistical maps. To this end, target functional volumes from each infant were registered to the target functional of infant 1, data set 3 (the infant and data set with the most useable data) using an affine transformation, with translation and rotation parameters subsequently hand-tuned. While these transformations were not perfect, insofar as linear registration cannot perfectly align different brains, they were primarily used for the registration of large search spaces, which should be tolerant to minor inaccuracies in registration. Lastly, to transform search spaces across infants and adults, this target functional volume was registered to the MNI brain using an affine transformation, with translation and rotation parameters subsequently hand-tuned.

**Data modelling.** For each pseudorun, whole-brain voxelwise linear models were performed to estimate the blood-oxygen-level-dependent response to visual stimuli. Regressors for each condition (excluding the baseline) were defined as boxcar functions with value 1 during blocks of that condition, convolved with a canonical double-gamma hemodynamic response function. Twelve nuisance regressors were additionally included to reduce the influence of potential artefacts. A linear trend regressor was included to account for signal drift. Motion parameter regressors (three translation parameters and three rotation parameters determined by motion correction) were used to minimize effects of head motion. Last, five principal component analysis (PCA)-based noise regressors were used to account for other noise sources (a method similar to GLMDenoise[39]). PCA-based regressors were defined by: (1) choosing a 'noise pool' of voxels with <1% of variance explained by the task regressors; (2) running PCA on time series from these voxels; and (3) choosing the top five principal components as regressors. For both task and nuisance regressors, time points that were scrubbed in data selection were removed after the regressors were defined (with the exception of PCA-based regressors, which were defined using scrubbed data).

This analysis provided beta values for task regressors corresponding to the magnitude of response to each condition, and contrast values corresponding to differences across conditions. To combine the resulting contrast values across pseudoruns for a given participant and data set, we computed a weighted average of contrast maps registered to a common functional space, using weights corresponding to the amount of data contributed by each pseudorun. Weights were proportional to $(c^T(X^TX)^{-1}c)^{-1}$, where $c$ is the contrast vector and $X$ is the design matrix for a given pseudorun. For a given contrast (for example, faces versus scenes), we combined data across all experiments containing that contrast.

We next statistically assessed these average contrast values for each participant. Because fMRI time series are temporally autocorrelated, within-participant statistics are typically computed using feasible generalized least squares, with an empirical estimate of the autocorrelation structure. However, the validity of extant methods for estimating the autocorrelation of fMRI data is not well established[40], and these methods have not been validated in infant data. To obviate the need for any assumptions about the autocorrelation structure in our data, we instead used a nonparametric permutation test[41]. Specifically, on each of 5,000 iterations, we randomly permuted the block order for each pseudorun, and computed a contrast value for each voxel. This procedure provided a null distribution that was used to threshold voxelwise contrast values at $P < 0.01$, one-tailed. Estimated null distributions were fit with a Gaussian distribution, allowing us to estimate small $P$ values that wouldn't be possible to estimate from the fraction of samples from the null distribution exceeding the observed statistic; for statistics with larger $P$ values, the Gaussian fit gave very similar $P$ values to those computed using the raw null distribution. For visualization and reporting purposes, voxelwise statistics were converted to $z$-values based on their computed $P$ value. To correct for multiple comparisons across voxels, we additionally used a permutation test to build a null distribution for sizes of contiguous clusters of activation, and thresholded cluster sizes at $P < 0.05$.

We additionally computed a group-level statistical map to perform inference across infants. Average contrast maps for each infant were registered to the target functional image of infant 1, data set 3, and voxelwise $t$-tests were performed across infants, comparing contrast values to zero, thresholded at $P < 0.01$. For infants with multiple data sets acquired at different ages, we only used the data set with the largest amount of saved data. As above, voxelwise $t$-statistics were converted to $z$-values based on their computed $P$ value for visualization purposes. To correct for multiple comparisons across voxels, a permutation test was used to build a null distribution for sizes of contiguous clusters of activation (where on each iteration, signs of contrast values for each infant were randomly flipped), and thresholded cluster sizes at $P < 0.05$.

**ROI analysis.** To assess response profiles of brain regions identified in the whole-brain analysis, we performed ROI analyses. ROIs were defined as the set of voxels within a broad anatomical search space with the top $N$% of statistical values for a specific contrast, such as comparing faces to scenes or faces to objects. The value $N$ was typically 5%, but was also varied from 2 to 30% to measure selectivity as a function of ROI size. Search spaces were hand-drawn on the anatomical image of one participant (infant 1, data set 3), and registered to other participants' functional images as described above (Data registration). They included (Supplementary Fig. 4): (1) lateral occipitotemporal cortex, covering the expected locations of the occipital face area and occipital place area (mean size 39.2 cm$^3$ in infants; 54.3 cm$^3$ in adults); (2) ventral temporal cortex, covering the expected locations of the fusiform face area and parahippocampal place area (38.4 cm$^3$ in infants; 54.0 cm$^3$ in adults); (3) STS, covering the expected location of the posterior STS face region (40.2 cm$^3$ in infants; 53.0 cm$^3$ in adults); and (4) medial prefrontal cortex (65.5 cm$^3$ in infants; 97.0 cm$^3$ in adults). To maximize the amount of data used to define regions, but still extract responses from data independent of those used to define the ROI[42], we used a leave-one-pseudorun-out analysis: ROIs were defined using data from all but one pseudorun, responses were extracted from the remaining pseudorun, and after iterating this process across all pseudoruns and participants, the resulting beta values and contrasts were combined using the weighted average described above (Data modelling). Beta values and contrasts were converted to per cent signal change values by dividing by mean signal strength within the ROI.

For most analyses, differences between conditions were statistically assessed using a permutation test, analogous to the procedure described above (Data modelling); these tests assess the significance of the observed effects within our sample. In addition, we tested whether the effects observed can be expected to generalize to the population. We compared responses to faces and scenes, because these conditions were observed by all infants, and combined data across all experiments to increase power within each participant. For each ROI (defined as described above, using the face versus scene contrast), mean per cent signal change values were computed for each participant, and the difference between responses to faces and scenes was statistically compared to zero using a one-tailed $t$-test across infants. As with the whole-brain group-level analysis, when infants yielded multiple data sets acquired at different ages, we only used the data set with the largest amount of usable data.

**Visual feature analysis.** We next asked whether responses in category-sensitive visual regions could be explained in terms of lower-level visual features. In particular, we focused on high- and low-frequency content and rectilinearity (the presence of 90° angles in an image), which have been argued previously to modulate responses in category-sensitive visual regions[24–26]. Frequency content and rectilinearity measures were computed on individual frames from each movie clip, and averaged across frames for a given clip. Frames were first converted to grayscale and normalized to have zero mean and unit standard deviation, to remove effects of overall luminance and contrast. We then computed the discrete Fourier transform of each frame, and defined low-frequency content as total power at frequencies less than one cycle per degree of visual angle, and high-frequency content as total power at frequencies greater than five cycles per degree of visual angle, following the cutoffs used by Razimehr et al.[25] Rectilinearity was computed using a procedure described by Nasr et al.[24]: frames were convolved with a bank of 90° angle Gabor filters at different scales and orientations, and magnitudes of convolved images were averaged across spatial position and filter to yield a single measure (Fig. 3).

We then assessed whether responses in category-sensitive ROIs were better predicted by category identity or by visual features. Regressors for visual features were defined by constructing time series of feature values for each individual movie in a given pseudorun, convolved with a canonical double-gamma hemodynamic response function. Categorical regressors were defined as described above (Data modelling). We compared five models: category (containing regressors for each visual category in an experiment), low-frequency content, high-frequency content, rectilinearity, and a model containing low-frequency, high-frequency and rectilinearity regressors. To eliminate the possibility that differences in model fit resulted from different degrees of freedom across models, model fit was assessed using leave-one-pseudorun-out cross-validation. For a given pseudorun, models were fit using data from all other pseudoruns with the same set of conditions from

that participant and data set (ROIs were also defined using data independent from the left-out pseudorun, as described in the ROI analysis section above). This provided a set of beta values that was used to define a predicted response for the left-out pseudorun, for each model. Model fit was assessed by computing the Fisher-transformed correlation ($z'$-value) between the time series in the left-out pseudorun and the predicted response. Linear trend and motion parameter nuisance regressors were included in all models. Model fit estimates were compared across models using paired, two-tailed $t$-tests across pseudoruns.

**Representational similarity analysis.** As an alternative method of comparing visual responses across infants and adults, we assessed the similarity structure of spatial patterns of response to different categories of stimuli[28]. Specifically, we computed correlations between spatial patterns of response (beta values comparing each condition to baseline) to the four conditions of Expts. 2–6, in infants and adults. Patterns were computed across voxels within extrastriate cortex, defined as the union of the three anatomical search spaces described above (ventral temporal cortex, lateral occipital cortex and the STS), with data combined across runs as described above (Data modelling). Correlation matrices (or representational similarity matrices, RSMs) were Fisher-transformed, averaged across participants and then inverse-Fisher-transformed for reporting.

To compare RSMs across groups, we next asked whether the ordering of correlation magnitudes across pairs or conditions (for example, face-object, face-body and so on) differed across infants and adults. We computed rank correlations (Kendall's tau) between correlation values from each pair of participants, either within infants, within adults, or between infants and adults, and asked whether orderings were more consistent (higher rank correlation) within group than between. To test whether the difference between within- and between-group rank correlations was significantly greater than zero, we performed an exact permutation test, building a null distribution for these values by computing them based on all possible group assignments of the six infants and three adults.

**Further paradigm details.** Across infants, eight slightly different experiments were run. Experiment 1 contained two categories (face and scene) and was run in every infant. Experiment 2 contained four categories (face, body, object and scene) and was run in a subset of $n = 4$ infants. Experiments 3–8 contained 3–4 categories and were each only run in a single infant. Experiments 3–7 used stimuli that are very similar to those used in Experiment 2, and were used in early scanning sessions before switching to Experiment 2. Experiment 8 contained distinct stimuli and was intended to provide additional evidence for generalization of category preferences across different specific videos. Because we did not acquire enough usable data with Experiments 3–8 to analyse them in isolation, they were ultimately only used in combination with other experiments, to increase power for various analyses. In particular, because all experiments contained face and scene categories, all were used for whole-brain face versus scene comparisons, and to define ROIs based on this contrast. Because Experiments 3–6 contained four categories, they additionally contributed to four-condition ROI responses.

Experiment 1 consisted of Filmed Faces and Baby Einstein Scenes conditions, as well as a baseline condition of spatially scrambled scenes (using $15 \times 15$ grid scrambling, as is the case for all scrambled stimuli). The Filmed Faces were 60 3 s-long close-up videos of children's faces on a black background, filmed by the experimenters, as used in a previous experiment in adults[43]. These videos did not contain parts of the body below the neck. The Baby Einstein Scenes were 36 3 s-long videos of scenes taken from the Baby Einstein video collection, which all depicted a three-dimensional (3D) spatial layout, and did not contain humans or animals.

Experiment 2 consisted of Filmed Front Faces, Filmed Objects*, Filmed Bodies, Filmed Scenes (presented at 80% size) and a baseline condition of spatially scrambled scenes (also presented at 80% size). The Filmed Front Faces were 30 3 s-long videos of front-view faces, similar to the Filmed Faces condition, but containing distinct specific videos. The Filmed Objects* were a set of 20 3 s-long close-up videos of children's toys on a black background (for example, rolling balls and moving gear toys), filmed by the experimenters. These 20 clips were selected from a larger set of 60 clips used in a previous experiment[43] (where the *denotes the subset), which were chosen to have virtually no information about 3D scene layout (for example, corners between walls or between a wall and a floor). The Filmed Bodies were a set of 60 3 s-long close-up videos of children's bodies or body parts (not showing faces) on a black background, as used in a previous experiment[43]. The Filmed Scenes were a set of 60 3 s-long videos filmed by the experimenters from a camera moving through an outdoor scene (for example, a road and a field), as used in a previous experiment[43]. These all depicted a 3D spatial layout, and did not contain humans or animals.

Experiment 3 consisted of Filmed Faces, Filmed Objects*, Filmed Bodies, Baby Einstein Scenes and a baseline condition of spatially scrambled scenes.

Experiment 4 consisted of Filmed Front Faces, Filmed Objects, Filmed Bodies, Filmed Scenes and a baseline condition of spatially scrambled objects. Filmed Objects were the full set of 60 filmed object videos from which the Filmed Objects* videos were selected.

Experiment 5 consisted of Filmed Faces, Filmed Objects, Filmed Bodies, Filmed Scenes and a baseline condition of spatially scrambled objects.

Experiment 6 consisted of Filmed Faces, Filmed Objects, Filmed Bodies, Baby Einstein Scenes and a baseline condition of spatially scrambled objects.

Experiment 7 consisted of Filmed Front Faces, Filmed Side Faces, Filmed Objects*, Baby Einstein Scenes (presented at 80% size) and a baseline condition of spatially scrambled scenes. The Filmed Side Faces were 35 3 s-long videos of side-view faces, similar to the Filmed Faces and Filmed Front Faces conditions but containing distinct specific videos.

Experiment 8 consisted of Baby Einstein Faces, Baby Einstein Objects, Animated Scenes and a baseline condition of spatially scrambled scenes. Baby Einstein Faces were three 18 s-long videos (containing multiple clips) of children's faces, taken from the Baby Einstein video collection. While these videos typically only contained faces, hands were occasionally presented in the vicinity of the face. Baby Einstein Objects were three 18 s-long videos (containing multiple clips) of children's toys and other objects in motion, taken from the Baby Einstein video collection. Animated Scenes were 18 6 s-long videos designed by having a camera move through an animated scene created using Blender 3D animation software. These all depicted a 3D spatial layout, and did not contain humans or animals.

**Data availability.** The stimuli, data and analysis code that support the findings of this study are available from the corresponding author on request.

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

## Acknowledgements

We thank the Packard Foundation, Ellison Medical Foundation and NSF (graduate research fellowship to B.D., and the Center for Brains, Minds and Machines, CCF-1231216 to N.K. and R.S.) for funding this research; Anna Wexler for assistance in stimulus creation; Grace Lisandrelli for assistance with recruitment and data collection; Jorie Koster-Hale, Bob Desimone, Charles Jennings and Winrich Freiwald for useful feedback on the manuscript; and all of our infants and parents for participating.

## Author contributions

B.D., N.K. and R.S. designed research; B.D., H.R., D.D.D. and R.S. collected data; B.K. and L.L.W. provided the infant head coil; A.T. provided technical assistance with data acquisition; B.D. and R.S. Analysed data; B.D., N.K. and R.S. wrote paper.

## Additional information

**Competing financial interests:** The authors declare no competing financial interests.

