## [Peer Review File · Nature Communications]

Reviewers' comments:

Reviewer #1 (Remarks to the Author):

Review of Deen et al., "Organization of high-level visual cortex in human infants"

This paper investigates cortical organization in very young (~3-8 month old) infants. The main research question is, if the pattern of responsivity in visual cortex is already similar to the patterns observed in adults, or only emerges at a later age. The results show that even at these early stages, extrastriate cortex is broadly organized, such that responses to the stimulus categories of faces and scenes are already biased towards the regions repeatedly identified in adults. These findings are important in determining the basis of visual development and cortical organization. In addition, the authors have overcome a number of significant technical challenges in attaining these results, and as such these results are highly valuable.

My main concern is that the statement "precise response profiles are subsequently refined through development" does not seem to be developed very well in the manuscript. While the evidence that cortical organization in infants and adults is already very similar is clear from the data, this refinement of responses seems to be less so.

Specifically, from the analyses as presented, it seems hard to infer if the differences between infants and adults, are really the result of refinement of cortical organization over development/time, or are based on other differences such as changes in size of cortical areas over time, or stimulus engagement. While the current manuscript goes some way towards this, it would be helpful to see this treated more fully in the manuscript. As the authors note, this question is important given the clear differences in infant and adult performance from both a behavioral and neuroanatomical (e.g. pruning, myelination) perspective.

In addition, there might be a logic/phrasing issue that I am having trouble with. The authors say: "Thus, while the large-scale spatial organization of responses to faces versus scenes is present in infants and remains a principal dimension of cortical organization into adulthood, highly selective regions for particular categories apparently emerge later in development, perhaps requiring more extensive visual experience."

However, these highly selective categories are derived by comparing responses to faces/scenes and other objects. It seems to me that faces vs. scenes could be considered to be at least as highly selective as other objects vs scenes.

Minor comments:

One typo:

P2: visual cortex ha-s- never been tested in infants

Reviewer #2 (Remarks to the Author):

Deen and colleagues report a technically impressive fMRI study of 4 - 6 month old human infants demonstrating adult-like, large-scale organization of category-specific responses to visual stimuli. Crucial technical advances include specially designed MRI receiver coils and experimental strategies for obtaining useable data in awake infants. Figs. 1 - 2 convincingly show face vs. scene specificity in infants in regions of the brain that are adult-like in topography. Fig. 3 demonstrates that regional category specificity (face, object, body-part, scene) is present in infants but is not as well developed as it is in adults. Statistical significance was assessed using permutation resampling thereby avoiding recently criticized commonly used parametric techniques (Eklund et al., 2016). These are all strong features. I see no reason to question the reported results. However, the scholarship is disappointing on two grounds.

First, the work is framed as relevant to a "debate" between two contrasting perspectives, one emphasizing that the infant brain is anatomically immature, the other pointing to evidence that response specificity is present early in life. There is no such "debate" as these perspectives are not mutually exclusive. A more correct framing would relate this work to the question of the extent to which brain development is experience dependent vs. independent, or, more precisely, the extent to which topographic specificity vs. functional specificity is experience dependent (Reid, 2012). Obviously, investigation of this question has been largely in experimental animals. Here, the data were obtained in humans. The reported results argue that an adult-like anatomical distribution of category specificity is present in the infant brain, which suggests that this feature of brain organization develops in an experience independent manner.

Second, relevant prior work in humans is not cited. It has recently been shown that topographically normal resting state functional connectivity is present within the visual system of humans who have never experienced sight (Striem-Amit et al., 2015). Clearly, that work is not about response specificity but it does demonstrate experience independent development of topographic specificity, hence, is pertinent. Similar results, reviewed in (Heimler et al., 2015), should have been cited here as well as in the senior author's recent paper on the development of the visual word form area (Saygin et al., 2016).

The distinction between topographic vs. response specificity has important clinical implications, as it helps to explain why useful sight cannot be restored by correction of refractive problems in humans who lacked visual experience during the critical period (Murray et al., 2015). Although discussion of results must be well focused in papers published in Nature journals, there still should be room for a broader sense of how the results illuminate important questions of general interest.

Minor point.

It would be somewhat more correct to cite (Siegel et al., 2014) for frame censoring of conventional fMRI. Power 2014 discusses frame censoring of resting state fMRI.

Eklund, A., Nichols, T.E., and Knutsson, H. (2016). Cluster failure: Why fMRI inferences for spatial extent have inflated false-positive rates. *Proc Natl Acad Sci U S A* 113, 7900-7905.

Heimler, B., Striem-Amit, E., and Amedi, A. (2015). Origins of task-specific sensory-independent organization in the visual and auditory brain: neuroscience evidence, open questions and clinical implications. *Curr Opin Neurobiol* 35, 169-177.

Murray, M.M., Matusz, P.J., and Amedi, A. (2015). Neuroplasticity: Unexpected Consequences of Early Blindness. *Curr Biol* 25, R998-R1001.

Reid, R.C. (2012). From functional architecture to functional connectomics. *Neuron* 75, 209-217.

Saygin, Z.M., Osher, D.E., Norton, E.S., Yousoufian, D.A., Beach, S.D., Feather, J., Gaab, N., Gabrieli, J.D., and Kanwisher, N. (2016). Connectivity precedes function in the development of the visual word form area. *Nat Neurosci* 19, 1250-1255.

Siegel, J.S., Power, J.D., Dubis, J.W., Vogel, A.C., Church, J.A., Schlaggar, B.L., and Petersen, S.E. (2014). Statistical improvements in functional magnetic resonance imaging analyses produced by censoring high-motion data points. *Hum Brain Mapp* 35, 1981-1996.

Striem-Amit, E., Ovadia-Caro, S., Caramazza, A., Margulies, D.S., Villringer, A., and Amedi, A. (2015). Functional connectivity of visual cortex in the blind follows retinotopic organization principles. *Brain* 138, 1679-1695.

Reviewer #3 (Remarks to the Author):

Category-sensitive visual cortex in human infants by Deen et al.

In this study, the authors scan a large number of human infants (17, of which 9 produced usable data) to explore the structure of infant ventral temporal cortex, and compare to that in adults. They use a variety of innovations to improve data quality of infant scans, including infant-sized MR head coils, quiet pulse sequences, and dynamic visual stimuli. In a comparison of activation to faces vs scenes, the infants (3-8 months) showed face- and scene-selective regions in locations matching those in adults. In this sense, the large-scale functional organization of infant ventral temporal cortex mirrors that in adults. However, there was no evidence for any category-selective regions, e.g., face-selective regions that responded more to faces than to any other category. Furthermore, within regions selective for face over scenes, a rectilinear model which simply predicted low quantity of rectilinear features could predict responses just as well as a category model. This shows that infant cortex is still undergoing changes (whether maturational or in response to experience, we still don't know), and can be considered only to have "proto" category selective cortex.

Overall, I found the manuscript very clearly written, and the methods and analyses appropriate. I strongly support publication. The present paper clearly shows that precursors of adult face and scene areas already exist at 3-8 months, but these regions are not yet fully adult-like in their selectivity and size

Reviewers' comments:

Reviewer #1 (Remarks to the Author):

Review of Deen et al., "Organization of high-level visual cortex in human infants"

This paper investigates cortical organization in very young (~3-8 month old) infants. The main research question is, if the pattern of responsivity in visual cortex is already similar to the patterns observed in adults, or only emerges at a later age. The results show that even at these early stages, extrastriate cortex is broadly organized, such that responses to the stimulus categories of faces and scenes are already biased towards the regions repeatedly identified in adults. These findings are important in determining the basis of visual development and cortical organization. In addition, the authors have overcome a number of significant technical challenges in attaining these results, and as such these results are highly valuable.

My main concern is that the statement "precise response profiles are subsequently refined through development" does not seem to be developed very well in the manuscript. While the evidence that cortical organization in infants and adults is already very similar is clear from the data, this refinement of responses seems to be less so.

Specifically, from the analyses as presented, it seems hard to infer if the differences between infants and adults, are really the result of refinement of cortical organization over development/time, or are based on other differences such as changes in size of cortical areas over time, or stimulus engagement. While the current manuscript goes some way towards this, it would be helpful to see this treated more fully in the manuscript. As the authors note, this question is important given the clear differences in infant and adult performance from both a behavioral and neuroanatomical (e.g. pruning, myelination) perspective.

We agree with the reviewer that the differences we find between infants and adults are important, but somewhat more complicated to interpret than the similarities between infants and adults (i.e. the surprisingly early spatial organization of functional responses). As described below, we have extended our discussion of this result to bolster our interpretation.

The reviewer asks whether the reported differences between infants and adults could result from changes in the size of functional regions. That is, could infants have selective regions that are just much smaller than the selective regions in adults? We conducted an analysis to test this (shown in Figure 3). To ask whether the lack of face>object or scene>object effects in infants depends on ROI size, we test ROIs varying in size from the top 2% to top 30% of voxels within an anatomical search space. We find that infants don't show either effect at any ROI size, while adults show both effects at all ROI sizes. We have added the following sentence to the main text to highlight this result (Results, paragraph 4):

Similar results were obtained for a range of ROI sizes (Fig. 3): adults showed a significant response to faces (or scenes) over objects for all regions and ROI sizes (permutation test, $n=3$, all P 's < .05), while infants did not show a significant response for any region or ROI size, including ROIs as small as $.8\text{cm}^3$ (permutation test, $n=6$, all P 's > .05). Thus, within the spatial

resolution of our methods, we find no evidence that the difference between groups reflects a change in the size of selective regions.

We have also reported the mean sizes of anatomical search spaces in the Methods section (Region-of-interest analysis subsection), so actual ROI sizes can be computed from percentages.

Regarding the possibility that the difference between infants and adults results from differences in stimulus engagement, we consider this account unlikely. For example, one might be concerned that infants were overall less engaged by our stimuli, yielding lower, less selective responses in all brain regions. However, as the contrast of face <> scenes shows, the infants were sufficiently engaged to produce robust category-sensitive cortical responses to both faces and scenes. In addition, although we found no regions that responded more to faces than objects (or scenes than objects), in the very same data we found a number of regions that were robustly recruited for objects more than faces (or scenes). Thus infants were sufficiently engaged by the object movies to elicit robust cortical responses to those movies. Overall, these results show that infants were engaged by movies of faces, scenes, and objects, but that no cortical regions responded more to faces than objects, or scenes than objects. The dramatic difference in response profile (compared to adults) can't plausibly be explained by differences in stimulus engagement.

That said, we appreciate the sentiment that one must be careful interpreting differences across infants and adults, given differences across these groups in various aspects of brain physiology and generic visual and cognitive function. We have added the following paragraph to the Discussion section to express this point:

Differences between infants and adults must be interpreted with caution, given the dramatic differences in brain size and general visual and cognitive function. For instance, one possibility is that in adults, category-selective responses are enhanced by top-down feedback and selective attention, which are not yet mature in infants. Nevertheless, these data are consistent with the hypothesis that the early-developing large-scale functional organization of category preferences in cortex provides a scaffolding for subsequent refinement of responses, leading ultimately to the strongly category-selective regions observed in adults.

In addition, there might be a logic/phrasing issue that I am having trouble with. The authors say: "Thus, while the large-scale spatial organization of responses to faces versus scenes is present in infants and remains a principal dimension of cortical organization into adulthood, highly selective regions for particular categories apparently emerge later in development, perhaps requiring more extensive visual experience."

However, these highly selective categories are derived by comparing responses to faces/scenes and other objects. It seems to me that faces vs. scenes could be considered to be at least as highly selective as other objects vs scenes.

We are using the term "selective" in this paper to refer to a region that has a substantially stronger response to its preferred category over any other category of visual input. Given this definition, finding a response to faces over scenes (or vice versa) is not sufficient to establish

selectivity, which requires testing responses to multiple categories. Methodologically, we use the contrasts of faces > objects and scenes > objects (rather than faces <> scenes) to search for selective regions simply because in adults, these contrasts are more effective at identifying regions with selective response profiles. As shown in Supplementary Figure 3, faces <> scenes is a relatively coarse contrast that yields responses in large swaths of cortex, whereas faces > objects and scenes > objects yield more focal, selective regions. The logic of our analysis is as follows: we don't find regions in infants that respond to faces (or scenes) over objects; a selective region (by our definition) would respond to faces (or scenes) over all other categories; hence we don't find selective regions in infants.

Minor comments:

One typo:

P2: visual cortex has- never been tested in infants

This typo has been corrected.

Reviewer #2 (Remarks to the Author):

Deen and colleagues report a technically impressive fMRI study of 4 - 6 month old human infants demonstrating adult-like, large-scale organization of category-specific responses to visual stimuli. Crucial technical advances include specially designed MRI receiver coils and experimental strategies for obtaining useable data in awake infants. Figs. 1 - 2 convincingly show face vs. scene specificity in infants in regions of the brain that are adult-like in topography. Fig. 3 demonstrates that regional category specificity (face, object, body-part, scene) is present in infants but is not as well developed as it is in adults. Statistical significance was assessed using permutation resampling thereby avoiding recently criticized commonly used parametric techniques (Eklund et al., 2016). These are all strong features. I see no reason to question the reported results. However, the scholarship is disappointing on two grounds.

First, the work is framed as relevant to a "debate" between two contrasting perspectives, one emphasizing that the infant brain is anatomically immature, the other pointing to evidence that response specificity is present early in life. There is no such "debate" as these perspectives are not mutually exclusive. A more correct framing would relate this work to the question of the extent to which brain development is experience dependent vs. independent, or, more precisely, the extent to which topographic specificity vs. functional specificity is experience dependent (Reid, 2012). Obviously, investigation of this question has been largely in experimental animals. Here, the data were obtained in humans. The reported results argue that an adult-like anatomical distribution of category specificity is present in the infant brain, which suggests that this feature of brain organization develops in an experience independent manner.

Second, relevant prior work in humans is not cited. It has recently been shown that topographically normal resting state functional connectivity is present within the visual system of humans who have never experienced sight (Striem-Amit et al., 2015). Clearly, that work is not about response specificity but it does demonstrate experience independent development of

topographic specificity, hence, is pertinent. Similar results, reviewed in (Heimler et al., 2015), should have been cited here as well as in the senior author's recent paper on the development of the visual word form area (Saygin et al., 2016).

The distinction between topographic vs. response specificity has important clinical implications, as it helps to explain why useful sight cannot be restored by correction of refractive problems in humans who lacked visual experience during the critical period (Murray et al., 2015). Although discussion of results must be well focused in papers published in Nature journals, there still should be room for a broader sense of how the results illuminate important questions of general interest.

We agree that one relevant theoretical debate addressed in this study is between experience-dependent and experience-independent accounts of visual cortical development, as stated in the introduction: “The origins of these responses have been the topic of intense debate: are they learned, reflecting a gradual accrual of expertise, or do they reflect innate predispositions?”

In order to better situate the current results in the context of this debate, and prior literature relevant to the debate, we have substantially extended the Discussion section of the paper, and added a number of references, including studies of category-sensitive responses in blind adults, and the Saygin, 2016 paper on the VWFA. The following new segments in the Discussion directly address the contribution of this paper to the question of dependence of visual development on experience:

Prior fMRI studies have observed category-sensitive responses in high-level visual cortex in children as young as 4 years⁸. By demonstrating that these responses exist by 4-6 months of age, the current study provides a stronger constraint on theories of cortical development: this functional organization must either be determined innately, without any need for visual experience, or develop within the first few months of life. A limited role for visual experience in the development of category-sensitive responses is consistent with evidence that in congenitally blind adults, category-sensitive responses in visual cortex develop in the absence of any visual input^{29, 30}.

...these data are consistent with the hypothesis that the early-developing large-scale functional organization of category preferences in cortex provides a scaffolding for subsequent refinement of responses, leading ultimately to the strongly category-selective regions observed in adults. The process of refinement likely depends on both physiological maturation (e.g. myelination of long-range connections between brain regions) and visual experience. For example, the visual word form area develops as a result of experience with a specific orthography³¹, but is guided by pre-existing patterns of anatomical connectivity³². Similarly, extensive training with novel symbols can generate selective responses in a cortical region in macaques; the location of this region is consistent across animals, suggesting refinement based on a pre-existing scaffold^{12, 13}.

Minor point.

It would be somewhat more correct to cite (Siegel et al., 2014) for frame censoring of conventional fMRI. Power 2014 discusses frame censoring of resting state fMRI.

We have replaced the Power, 2014 reference with the suggested Siegel, 2014 reference, as well as the original Power, 2012 article, on the basis of historical precedence.

Eklund, A., Nichols, T.E., and Knutsson, H. (2016). Cluster failure: Why fMRI inferences for spatial extent have inflated false-positive rates. *Proc Natl Acad Sci U S A* 113, 7900-7905.

Heimler, B., Striem-Amit, E., and Amedi, A. (2015). Origins of task-specific sensory-independent organization in the visual and auditory brain: neuroscience evidence, open questions and clinical implications. *Curr Opin Neurobiol* 35, 169-177.

Murray, M.M., Matusz, P.J., and Amedi, A. (2015). Neuroplasticity: Unexpected Consequences of Early Blindness. *Curr Biol* 25, R998-R1001.

Reid, R.C. (2012). From functional architecture to functional connectomics. *Neuron* 75, 209-217.

Saygin, Z.M., Osher, D.E., Norton, E.S., Youssoufian, D.A., Beach, S.D., Feather, J., Gaab, N., Gabrieli, J.D., and Kanwisher, N. (2016). Connectivity precedes function in the development of the visual word form area. *Nat Neurosci* 19, 1250-1255.

Siegel, J.S., Power, J.D., Dubis, J.W., Vogel, A.C., Church, J.A., Schlaggar, B.L., and Petersen, S.E. (2014). Statistical improvements in functional magnetic resonance imaging analyses produced by censoring high-motion data points. *Hum Brain Mapp* 35, 1981-1996.

Striem-Amit, E., Ovadia-Caro, S., Caramazza, A., Margulies, D.S., Villringer, A., and Amedi, A. (2015). Functional connectivity of visual cortex in the blind follows retinotopic organization principles. *Brain* 138, 1679-1695.

Reviewer #3 (Remarks to the Author):

Category-sensitive visual cortex in human infants by Deen et al.

In this study, the authors scan a large number of human infants (17, of which 9 produced usable data) to explore the structure of infant ventral temporal cortex, and compare to that in adults. They use a variety of innovations to improve data quality of infant scans, including infant-sized MR head coils, quiet pulse sequences, and dynamic visual stimuli. In a comparison of activation to faces vs scenes, the infants (3-8 months) showed face- and scene-selective regions in locations matching those in adults. In this sense, the large-scale functional organization of infant ventral temporal cortex mirrors that in adults. However, there was no evidence for any category-selective regions, e.g., face-selective regions that responded more to faces than to any other category. Furthermore, within regions selective for face over scenes, a rectilinear model which simply predicted low quantity of rectilinear features could predict responses just as well as a category model. This shows that infant cortex is still undergoing changes (whether maturational or in response to experience, we still don't know), and can be considered only to have "proto" category selective cortex.

Overall, I found the manuscript very clearly written, and the methods and analyses appropriate. I strongly support publication. The present paper clearly shows that precursors of adult face and scene areas already exist at 3-8 months, but these regions are not yet fully adult-like in their selectivity and size

REVIEWERS' COMMENTS:

Reviewer #1:

I thank the authors for their considered response and resultant changes to the manuscript. They have satisfactorily addressed my prior concerns.

Reviewer #2:

This work is technically excellent. The Discussion section now much more fully relates the present findings to the developmental literature. Space limitations preclude citing all relevant material. Accordingly, I recommend R. C. Reid's article in Neuron 2012 (doi: 10.1016/j.neuron.2012.06.031) for additional discussion of the distinction between topographic vs. functional specificity.

A. Z. Snyder

Reviewers' comments:

Reviewer #1:

I thank the authors for their considered response and resultant changes to the manuscript. They have satisfactorily addressed my prior concerns.

Reviewer #2:

This work is technically excellent. The Discussion section now much more fully relates the present findings to the developmental literature. Space limitations preclude citing all relevant material. Accordingly, I recommend R. C. Reid's article in *Neuron* 2012 (doi: 10.1016/j.neuron.2012.06.031) for additional discussion of the distinction between topographic vs. functional specificity.

A. Z. Snyder

We have added this reference to the Discussion (Ref. 31, paragraph 4), while discussing the hypothesis that spatial organization develops early and guides subsequent development of functional specificity.